immunology/microbiology/health and disease and epidemiology

Group A Streptococcus, gas vaccine, rheumatic fever, pharyngitis, toxic shock syndrome, M protein

**Author for correspondence:**
Helge C. Dorfmueller
e-mail: h.c.z.dorfmueller@dundee.ac.uk

# A brief review on Group A Streptococcus pathogenesis and vaccine development

Sowmya Ajay Castro and Helge C. Dorfmueller

Division of Molecular Microbiology, School of Life Sciences, University of Dundee, Dow Street, Dundee, DD1 5EH, UK

SAC, 0000-0001-9831-2907; HCD, 0000-0003-1288-044X

*Streptococcus pyogenes*, also known as Group A Streptococcus (GAS), is a Gram-positive human-exclusive pathogen, responsible for more than 500 000 deaths annually worldwide. Upon infection, GAS commonly triggers mild symptoms such as pharyngitis, pyoderma and fever. However, recurrent infections or prolonged exposure to GAS might lead to life-threatening conditions. Necrotizing fasciitis, streptococcal toxic shock syndrome and post-immune mediated diseases, such as poststreptococcal glomerulonephritis, acute rheumatic fever and rheumatic heart disease, contribute to very high mortality rates in non-industrialized countries. Though an initial reduction in GAS infections was observed in high-income countries, global outbreaks of GAS, causing rheumatic fever and acute poststreptococcal glomerulonephritis, have been reported over the last decade. At the same time, our understanding of GAS pathogenesis and transmission has vastly increased, with detailed insight into the various stages of infection, beginning with adhesion, colonization and evasion of the host immune system. Despite deeper knowledge of the impact of GAS on the human body, the development of a successful vaccine for prophylaxis of GAS remains outstanding. In this review, we discuss the challenges involved in identifying a universal GAS vaccine and describe several potential vaccine candidates that we believe warrant pursuit.

## 1. Introduction

Group A Streptococcus (GAS) or *Streptococcus pyogenes* is a virulent Gram-positive pathogen responsible for a plethora of diseases ranging from mild, superficial infections to life-threatening diseases with high morbidity and mortality in humans [1]. GAS is responsible for causing around 700 million cases of pharyngitis annually worldwide. Increasing incidence of mild symptoms such as strep throat may lead to an invasive prevalence of conditions such as necrotizing fasciitis (NF; also called flesh-eating disease), streptococcal toxic shock syndrome

(STSS) and other post-infectious immune-related diseases at the population level [1]. Acute rheumatic fever (ARF), triggered by an autoimmune response following GAS infection, is one of the major causes of rheumatic heart disease (RHD) leading to high mortality rates in non-industrialized countries [2].

GAS remains globally sensitive to penicillin, despite reports that penicillin has failed to eradicate GAS pharyngitis and tonsillitis [2–5]. Importantly, GAS isolates remain susceptible to other β-lactam antibiotics such as amoxicillin and cephalosporins [6]. Other commonly used antibiotics to treat GAS infections, in situations of penicillin allergy, are clindamycin and the macrolides. However, resistance against both these alternatives has been reported [7]. Strikingly, a recent study demonstrated that a rare mutation, found in the penicillin-binding protein 2B in two GAS strains, confers reduced susceptibility to β-lactam antibiotics from the penicillin family, including amoxicillin [8]. This very same initial mutation occurred in *Streptococcus pneumoniae* and eventually led to penicillin resistance [9]. It is alarming to learn that GAS could be on the path to becoming resistant to the most frequently prescribed antibiotics, including penicillin and amoxicillin. The most promising approach to combat future antibiotic resistance mechanisms would be a GAS vaccine.

The main challenge in identifying an effective and safe GAS vaccine that has remained unchanged for decades is the production of a universal vaccine candidate to protect us from extant and emerging GAS strains. A recent genomic study reported by Davies *et al.* [10] analysed a database of more than 2000 publicly available GAS genomes. Of these isolates, 649 GAS were described as newly emerged clones. This extensive genomic study reported 13 possible antigenic proteins as being conserved in over 99% of isolates found globally.

Historically, the most common possible antigenic targets of GAS were divided into two categories: (i) M protein-based candidates and (ii) non-M protein-based candidates. The M protein, encoded by the *emm* gene, is an immunodominant GAS protein, consisting of a coiled-coil structure, which is deposited on the surface of the bacterial cell wall. Almost all clinical GAS strains are differentiated by the presence of their M proteins. The M proteins are widely researched for their ability to adhere to host cells and block phagocytosis, thereby assisting GAS colonization [11]. Today, more than 250 different M proteins are known, and their sequence variations make it challenging to find conserved protein domains/motifs that are present in most GAS serotypes. The M protein is structurally (schematic structure of M-proteins are detailed in [12–14]) and functionally a versatile protein. The M-protein interaction with the host is reviewed in [15]. Recent investigations of potential GAS vaccine candidates resulted in the development of a vaccine formulation containing either the N-terminal or C-terminal domains of selected M proteins, or a combination of both, to analyse the protective efficacy against GAS infections [16]. A brief review of the current progress of multivalent M proteins and non-M protein vaccine candidates was recently published [17].

However, a major problem for GAS vaccine development is antibody cross-reactivity with human organs, in particular with the myosin proteins in heart tissue [18]. The first evidence of cross-reactivity between anti-streptococcal antibodies and human heart tissues was found in mice immunized with GAS components [19]. Conversely, GAS pathogens were well recognized by antibodies produced against human heart extracts. In addition to myosin, skeletal myosin, tropomyosin, keratin, vimentin and laminin were also identified as cross-reactive host tissue proteins. Evidence shows that M protein is the strongest candidate to react with heart proteins. Non-M proteins such as N-acetyl-β-D-glucosamine, the immunodominant epitope of the Group A surface carbohydrate, hyaluronic capsule and two proteins (60 and 67 kDa) present in the GAS cell membrane, were also identified as cross-reactive antigens [20–23]. Hence, M proteins are not considered as effective vaccine candidates, unless human tissue cross-reactivity can be eliminated.

In addition to the M proteins as vaccine candidate antigens, numerous research groups are investigating non-M protein antigens for their efficacy and safety as vaccine candidates (figure 1). To date, validated candidate vaccine antigens are the proteins: streptococcal pyrogenic exotoxin (Spe), fibronectin-binding proteins (FBI), *Streptococcus pyogenes* cell envelope proteinase (SpyCEP), streptococcal C5a peptidase (ScpA) and streptolysin O (SLO) [24–28]. The only non-protein GAS vaccine candidate antigen is a carbohydrate component of the ubiquitous surface-exposed Group A Carbohydrate (GAC) [29]. Furthermore, a combination of several GAS antigens has also been developed as a 'multi-component vaccine' and has shown to induce protective efficacy in animal GAS infection models [16]. However, there are drawbacks to the use of non-M protein GAS antigens such as the high level of sequence variation and low sequence coverage across global GAS strains. In addition, these protein vaccine targets must be present in all strains to achieve protection against all GAS isolates. It will also be important to eliminate any autoimmune responses that could lead to

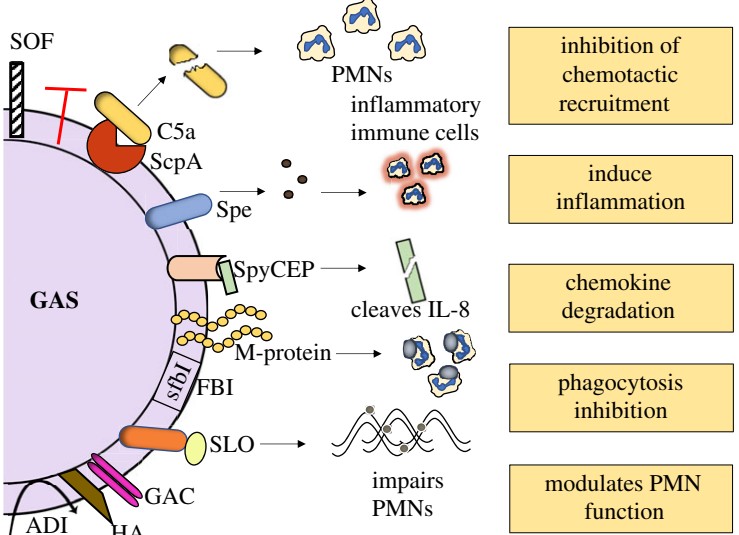

**Figure 1.** Virulence factors of GAS. A variety of antigens on the surface of the GAS are involved in virulence. Each of the displayed antigens have been well documented for their association to impair the host immune system. GAS produces several secreted toxins that cleave human proteins. Examples are ScpA, which cleaves the chemoattractant C5a and spyCEP cleaves neutrophil attracting chemokines, e.g. IL-8 on PMNs. This in turn inhibits the phagocyte recruitment. M-proteins bind to the components of the immune system thereby conferring resistance to phagocytosis. SLO impairs neutrophil function, whereas the carbohydrates GAC and HA promote GAS survival within the human blood. Abbreviations: NETs – Neutrophil extracellular traps; PMNs – Polymorphonuclear leukocytes; ScpA, streptococcal C5a peptidase; Spe, streptococcal pyrogenic exotoxin; SpeA, streptococcal pyrogenic exotoxin A; spyCEP, streptococcal pyogenes cell envelope protease; GAC, Group A Carbohydrate; FBI, fibronectin-binding protein; sfbl, *S. pyogenes* fibronectin-binding protein; SOF, serum opacity factor; ADI, arginine deaminase; HA, hyaluronic acid capsule; GAS, Group A Streptococcus.

autoimmune sequelae, e.g. the association between the N-acetylglucosamine (GlcNAc) side chain of GAC and ARF [29].

This review focuses on (i) outlining pathogenic mechanisms of GAS, (ii) the challenges of developing a universal GAS vaccine, and (iii) the vaccine candidates currently being developed to prevent GAS infections.

# 2. Group A Streptococcus is an obligate human pathogen

Unlike groups B, C and G streptococci, which are human and veterinary pathogens, the only natural reservoir for GAS is humans. The life cycle and diseases caused by GAS have not been reported to occur naturally in animals: perhaps other environmental reservoirs and biological causes remain to be discovered. Interestingly, GAS has sporadically been recovered from environmental sources including canine faeces and conjunctive discharge, and also found in association with a wild European hedgehog [30–32]. Although humans are the exclusive biological host for this pathogen, GAS infections can be mimicked in non-human primates as an infection model. For instance, unnaturally high doses of M1T1 GAS are able to colonize and induce pharyngitis and tonsillitis in Indian rhesus monkeys [33]. Other models such as C57BL/6, BALB/c, FVB/NJ mice and rabbits have been used as infection models to understand GAS pathogenesis [29,34].

## 2.1. Group A Streptococcus adhesion and colonization

The primary spread of GAS is through person-to-person transmission of contaminated air droplets [1]. GAS survives on the skin, and inside the host, for several hours to days [35], and a wide range of surface GAS proteins contributes to the attachment of the pathogen to the skin. Several GAS components, including the hyaluronic acid capsule (HA), fimbrious structures or pili (long rod-like structures that protrude from the surface of the bacterial cell wall), M proteins and the *S. pyogenes* fibronectin-binding adhesin (SfbI), contribute to adhesion and colonization of the pathogen in the nasopharynx region including tonsil epithelium and skin [36].

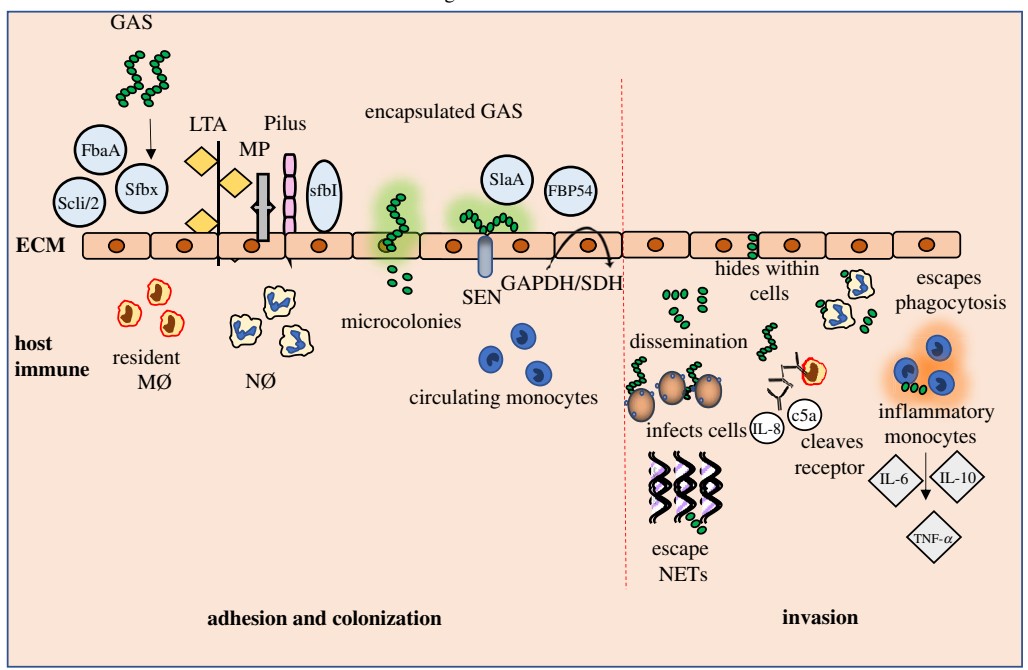

**Figure 2.** Stages of GAS invasion of the host immune system. A wide range of bacterial protein adhesins engage with the adherence and colonization of the GAS pathogen to the ECM of the host tissue. Initial attachment of GAS is followed by formation of microcolonies accompanied by cell wall-anchored adhesins and anchorless enzymes. Once colonized within the tissue sites GAS disseminates inside the host by surviving and multiplying. GAS survives by different mechanisms, including hiding within the epithelial cell lines, inhibiting phagocytosis and degrading DNase of NETs. GAS-infected cells trigger a strong inflammatory response, thereby inducing a cytokine storm. Abbreviations: GAS, Group A Streptococcus; ECM, extracellular matrix; LTA, lipoteichoic acid; MP, M-protein; FbaA, Scli/2, sfbX, sfbI, SlaA, FBP54, protein adhesins; SEN, streptococcal surface enolase; streptococcal surface dehydrogenase, GAPDH/SDH; MØ, macrophages; NØ, neutrophils; NETs, neutrophil extracellular traps.

While cell attachment is a highly complex event that is yet to be fully characterized, it is currently described as a two-step process. Initial attachment is facilitated by the GAS surface carbohydrate, lipoteichoic acid, which has weak but sufficient affinity to pharyngeal or dermal epithelial cells of the host through hydrophobic interactions [37]. Later stages are initiated by the high-affinity binding events, initiated by pili, and subsequent affinity via lectin–carbohydrate and protein–protein interactions. These interactions are mediated by GAS adhesion proteins and generate firm adhesion to distinct tissue sites in the human host [37]. Bacterial adherence is considered a dynamic process due to the ability of the pathogen to detach from the tight adhesion surface to transfer to a more favourable environment where they can survive and multiply [37,38]. Numerous human extracellular proteins such as collagen, fibronectin, fibrinogen, laminin and vitronectin serve as binding components for the GAS adhesins. Importantly, human fibronectin is a frequent binding site for the streptococcal adhesins, thus contributing to specific binding affinities between GAS and host $\alpha5\beta1$ integrin receptors on epithelial cells. GAS strains express at least 11 fibronectin-binding adhesins, including several M proteins, and they bind to host fibronectin either in a soluble or immobilized form in the extracellular matrix (ECM) [36]. Once GAS attaches to the host skin or pharynx surface, microcolony formation occurs, appearing as macroscopic structures that multiply and subsequently cause streptococcal infections (figure 2).

The upper respiratory tract favours a rich environment for the growth of many pathogens. GAS competes with the epithelial lining of the respiratory tract to colonize and invade the host epithelial cells. Several GAS strains have been reported to penetrate the intracellular space and survive within respiratory cells for 4 to 7 days [39]. The first observation of colonization of GAS was recorded in 1991 by Wessels et al., revealing that encapsulated GAS strains perform better than non-encapsulated strains at colonization [40]. Evidence supports that long-term colonization of GAS strains results in a frameshift mutation in the hasA gene that encodes HA capsule biosynthesis, contributing to either reduced, or absent, capsule production [41]. The HA capsule acts as a barrier to prevent phagocytosis by the innate immune cells. The HA also functions as a non-protein adhesin by binding to skin and

murine epithelial keratinocytes mediated through CD44, a HA receptor expressed on the surface of keratinocytes [42].

Genes of the component regulatory system (control of virulence), multiple gene regulator and RofA-like proteins are involved in the control of the expression of colonization function [43]. Adhesion and colonization of GAS are tightly regulated, multiple-level processes that are extensively reviewed in [37,44].

## 2.2. Host cell invasion

The invasion of GAS into human epithelial non-phagocytic cells, appearing as an intracellular bacterium, was first demonstrated by LaPenta et al. in 1994 [45]. The frequency of GAS invasion into human epithelial cells is similar to that of other classical intracellular pathogens such as Salmonella and Listeria [46]. Subsequent work by Österlund and colleagues described the use of immunohistological methods to stain and visualize the GAS cells in surgically excised tonsils from patients with frequent episodes of GAS [39,47,48]. Several studies have investigated the contribution and location of streptococcal invasin proteins and revealed a promotion of actin rearrangement of the host cell cytoskeleton. SfbI and M proteins are the best-studied GAS invasins that contribute to the invasion of epithelial, endothelial and phagocytic cells, respectively [36]. Intracellular dissemination of GAS occurs once the pathogen is in contact with the human epithelial cells. Most of the GAS pathogens are then engulfed; however, one of the vital characteristics of GAS is to 'hide' within a human cell, or at the endothelial barrier, leaving the host asymptomatic for a few days post-invasion [44]. This 'hiding' allows the pathogen to spread and colonize the site, e.g. the throat, and initiate the occurrence of mild symptoms such as strep throat.

Depending on the type of streptococcal invasin protein, the invasion of GAS into epithelial cells varies: a prominent example being SfbI attachment to human ECM. Human fibronectin serves as a platform for the binding of GAS SfbI to the host ECM. Traditionally, it was believed that GAS binding to integrin receptors triggered caveolae aggregation to form large invaginations in the host epithelial cells, which aid in taking up GAS. Alternatively, M protein binding to fibronectin generates a zipper-like structure, which in turn triggers GAS elimination from the epithelial cells through phagolysosomes [11]. However, the use of caveolae by GAS for uptake into epithelial cells is not conclusive. Recent work by Lim et al., shows that caveolae, and their scaffolding protein CAV1, protect the human epithelial HEp-2 cells against GAS invasion by a caveolae-independent mechanism. Knockout studies further demonstrated that CAV1 protein extends the host protection to the SfbI-expressing M12 strains [49]. This indicates that, regardless of SfbI expression, CAV1 proteins do not mediate the uptake of GAS into HEp-2 epithelial cells. Moreover, electron microscopy studies show M1T1[5448] GAS uptake by the plasma membrane invaginations possessing putative actin filaments, confirming that GAS invasion promotes reorganization of the host actin cytoskeleton [36,49].

Strikingly, GAS survival was shown within professional phagocytes, such as macrophages, by blocking armaments within the cell. This was first shown by Molinari & Chhatwal [50], demonstrating that GAS isolates escape phagolysosomes to multiply within the host cell cytoplasm [50]. These escape mechanism of GAS were also shown in neutrophils [51]. Soft tissue biopsies from patients with GAS showed the presence of live pathogens inside the macrophages. A correlation of bacterial numbers at the infected site was observed: low numbers of GAS were found within non-inflamed tissues, while high GAS numbers correlated with highly inflamed tissue, indicating that severity of infection tracks colony numbers [52]. It was shown that GAS pathogen survival is linked to streptolysin O that promotes escape from GAS-containing vacuoles into the macrophage cytosol [53]. This illustrates that GAS has potential to invade and escape phagolysosome and survival within the host cytoplasm.

Recent advances in research on the transmission of GAS have shown that S. pyogenes can access the bloodstream of the host through the lymphatic system. This study, conducted using an established animal soft tissue GAS invasion model, found that S. pyogenes disseminates via afferent and efferent lymphatics to reach local and distant lymph nodes, respectively, while remaining as an extracellular pathogen [54]. Although earlier work had indeed noted that bacteria were capable of passing through the lymphatic system [55], this latest work has provided a new perspective on GAS invasion and the development of streptococcal pathogenesis [54].

## 2.3. Non-invasive Group A Streptococcus infections

Early signs of non-invasive GAS infection include pharyngitis, impetigo, rash and redness. However, the first most common clinical symptom of GAS infection is the onset of pharyngitis. Several M protein serotypes, such as M1, M3, M5, M6, M14, M18, M19 and M24, were found to be associated with

pharyngitis and ARF. These serotypes fail to express the serum opacity factor (SOF), a virulent agent of GAS. Nevertheless, serotypes that express SOF, such as M2, M49, M57, M59, M60 and M61, are commonly associated with pyoderma and acute glomerulonephritis [56,57]. Moreover, depending on the season, the prevalence of different disease manifestations of GAS infection varies. For instance, pharyngitis and ARF have high incidence rates during autumn and winter, whereas skin infections and pyoderma commonly occur during the summer [1].

Specific GAS virulence factors are known to cause specific GAS symptoms. For example, GAS strains that produce streptococcal superantigens, such as the streptococcal pyrogenic exotoxins, generally cause symptoms including rash, 'strawberry' tongue, swollen glands and high temperature, as seen in scarlet fever in young children. An outbreak of scarlet fever in mainland China identified serotype M12 association with the streptococcal superantigen gene, streptococcal pyrogenic exotoxin C and the DNase gene spd1 [58]. A recent study by Lynskey et al. [59] has revealed that a new M1 subtype [M1$_{UK}$] is responsible for scarlet fever cases in the UK and exhibits higher levels of streptococcal pyrogenic exotoxins type A (SpeA). Subsequent studies by Rümke et al. [60] revealed that M1$_{UK}$ has also been a dominant clade in the Dutch population and is predicted to be present in many other populations.

## 2.4. Post-infection immune sequalae

Recurrent GAS infections are directly linked to life-threatening conditions such as STSS, NF and post-streptococcal glomerulonephritis (PSGN). Necrotizing fasciitis is a life-threatening aggressive disease caused by the invasion of GAS and destruction of the soft tissue, leading to high mortality rates and long-term morbidity worldwide. While many underlying details remain to be explored, animal models challenged with the wild-type M3 strain, which commonly causes NF in humans, showed extensive development of myonecrosis. By contrast, animals challenged with mutant M3 strains, deficient in either the HA capsule or the M protein, developed abscesses but no soft tissue destruction [61]. This suggests that the HA capsule, which is present in highly encapsulated GAS strains, might contribute to the deadly NF. However, recent outbreak strains M89 and M4, which do not produce HA capsule, have also been linked to invasive disease, thus implicating the involvement of other GAS toxins [62]. Increased expression of NADase and streptolysin O were found in the new clade emm89 variant, an acapsular strain that might be capable of causing invasive infections [62].

Further studies on GAS-induced NF demonstrated that GAS hijacks neuronal regulation by secreting streptolysin S, which activates the nociceptor (or 'pain receptor'). Activation of nociceptor releases enormous doses of neuropeptides that block the recruitment of neutrophils and inhibit further uptake and phagocytosis of GAS. In the mouse model, botulinum neurotoxin A, from the bacterium *Clostridium botulinum*, targets the peripheral nervous system by silencing the nerve fibres and thereby preventing GAS-induced NF [63].

First documented in the 1980s, STSS was observed in the Western countries, including the United States and Europe. Clinically, 85% of patients with STSS were infected with GAS strains that had increased SpeA expression [64]. Moreover, rabbits inoculated with purified SpeA protein displayed high temperature, hypotension and multiple organ failure: all classic symptoms of STSS [65]. A few studies have noted a link between the M1, M3 strains and SpeA expression and STSS, which is observed as a correlation rather causation [66,67]. The whole-genome sequence of the M3 strain isolated from an STSS patient revealed extensive chromosomal rearrangement and genetic variation compared with M1 and M18 strains, providing new insights into the molecular pathogenesis of STSS [68]. The contribution of SpeA to GAS invasive diseases is described in the section below: Group A Streptococcus vaccine candidates in the pre-clinical phase.

One of the widely accepted immediate therapies for STSS is intravenous immunoglobulins (IVIG). The pooled IgG contains neutralizing antibodies, which inhibit T cell proliferation and pro-inflammatory production triggered by streptococcal superantigens. Additionally, the IVIG study reported that infusion of immunoglobulins neutralized the streptococcal superantigenic activity in severe invasive GAS patients, including those with STSS and NF [69]. This suggests that investigating the antigenic targets found in IVIG could potentially be a novel route to develop GAS vaccine candidates [70].

## 2.5. Autoimmune diseases

GAS has been associated mainly with post-infection, immune-related, sequelae such as ARF and acute glomerulonephritis. ARF, a major cause of RHD in non-industrialized countries, occurs primarily in children and young adults and leads to disability or death. Although the host immune and

autoimmune responses are involved in causing rheumatic disease, the primary initiating factor is GAS infection [71]. The pathogenic mechanism of ARF is yet to be explored in detail, but immunopathology studies show that the onset of GAS adhesion and invasion triggers pharyngitis in the host. This is followed by the processing of GAS antigens and their presentation to B and T cells, leading to robust stimulation of cross-reactive monoclonal antibodies. The excreted antibodies mediate tissue injury, leading to the development of the classical features of ARF, including carditis [71]. It was also shown that anti-streptococcal antibodies, specifically anti-streptolysin O and anti-DNase antibodies, were recovered from patients with recurrent GAS infections, implying that the occurrence of ARF correlates with the onset of streptococcal infections [72]. Although only a small percentage of the population with streptococcal pharyngitis eventually develops ARF, studies conducted within families with ARF suggest that genetic and environmental factors also play a major role in determining the occurrence of this disease [73].

Post-streptococcal glomerular nephritis (PSGN) is another common autoimmune disease usually initiated by GAS pharyngitis, impetigo and scarlet fever. Frequent GAS episodes trigger the body's immune response of releasing cytokines to fight the pathogen, resulting in PSGN. Serotypes M1, M2, M4 and M12 have been linked with nephritis [74]. However, other streptococcal groups, such as Group C streptococci, have also been found contributing to acute PSGN. As well as M proteins, enolase and the V1 region in streptokinase (GAS-secreted enzymes) are found in systemic rheumatic disease and PSGN [74,75].

Outbreaks of GAS have been reported in many countries, including China in 2011 with a spike in scarlet fever [76], in a nursing facility in the United States between 2014 and 2016 with multiple invasive GAS infections [77], and more recently, in 2019, Public Health England reported invasive GAS cases in the United Kingdom [78]. To tackle these types of outbreaks and prevent future occurrences, an effective vaccine is essential.

# 3. Why is it challenging to develop a universal Group A Streptococcus vaccine?

Developing a vaccine that offers coverage for all globally identified GAS serotypes plus any future strains deriving from them is very challenging. Whole GAS genome sequencing has revealed two major issues: (i) an extensive genomic heterogenicity of GAS isolates due to frequent genetic recombination events, including gene exchange and single nucleotide polymorphisms, and (ii) subsequent protein sequence variations. No single protein that has been exploited for vaccine development so far, is 100% conserved throughout all GAS isolates [10]. Hence, the challenges that require to be addressed to deliver a universal GAS vaccine include (i) the attributed burden of GAS disease, (ii) safety, and (iii) diversity and antigenic variation in GAS strains (electronic supplementary material, figure S1).

## 3.1. Global burden

A deeper investigation of public health and, in particular, communicable invasive GAS diseases reveals that around 34 million people are affected worldwide per annum, with more than 10 million patients left permanently disabled [79]. Most cases affect children and young adults from low- and middle-income countries. Extensive knowledge and deeper research on GAS pathogenesis over the last few decades encourages an urgent need to develop a vaccine. The WHO lists GAS as the ninth infectious disease affecting people worldwide and has subsequently flagged developing a safe and global vaccine against GAS as a priority in their agenda [79]. However, gaps still remain in our current knowledge, hindering the development of vaccines, including both scientific evaluation and the lack of statistical data on current rates of global infections, which delay the accuracy of predicting the current GAS global burden [79].

## 3.2. Safety considerations

GAS vaccination human trials in the 1960s, using a crude extract of M protein from type 3 GAS, resulted in devastating side effects in the volunteers, 12.5% of whom developed ARF (3 out of 21 volunteers) [80]. This vaccine trial prompted the US Food and Drug Administration to forbid the use of GAS bacteria and their products as vaccine candidates. However, a workshop, conducted by the US National Institute of Allergy and Infectious Diseases in 2004, was instrumental in revoking the ban with the revision that

purified GAS antigens could be used as vaccine candidates [81,82]. Up to the present, approximately three dozen GAS antigens have been evaluated as protective and effective vaccine candidates in pre-clinical studies. Due to adverse effects, and the limitations of using animal models to evaluate GAS vaccine efficiency and safety for humans, several vaccine candidates have not been carried forward into clinical trials. Importantly, a number of vaccine candidates, such as the M protein domains, trigger a promising immune response by reducing the clinical symptoms in the animal models, but concerns remain regarding cross-reactivity with cardiac proteins [79]. Due diligence is needed to monitor post-vaccination studies by screening serum for cross-reactivity against human tissue and proteins.

## 3.3. Diverse Group A Streptococcus strains and antigenic variation

The global distribution of *emm* types in GAS is extremely diverse. Davies *et al*. [10] compiled a database of the GAS genome sequences of 645 geographically and clinically diverse strains. The database comprises 150 *emm* types which are clustered into 39 M proteins according to their protein sequences. This detailed analysis reveals protein sequence variations and heterogenicity within several current GAS vaccine candidates, including the protein streptolysin O and C5a peptidase. The data analysis also revealed a high level of gene plasticity throughout the more than 2000 GAS genomes, implying high-sequence diversity within the existing strains. GAS diversity varies depending on the global region: for example, *emm*1 and *emm*12 serotypes are observed less often, but with greater strain diversity, in Africa and the Pacific regions, whereas high occurrence with less strain diversity is recorded in South Asia and Latin American countries [10]. Tartof *et al*. argue that strain diversity depends on the social determinants present in the affected areas. For example, greater strain diversity was noted in the poorest regions than in the high-income suburbs within Brazil [83]. This observation poses challenges to developing a broad coverage GAS vaccine that benefits both the high-burden regions and the high-income countries.

As mentioned above, pre-clinical studies have shown that 28 vaccine candidates are protective against GAS infections using various animal models. However, the sequence conservation of the antigens throughout GAS strains was not determined. Davies *et al*. conducted a comprehensive analysis and revealed that the sequence variation in the antigen expression modulates the regulation of the host immune response. The authors provide a tool to aid vaccine antigen selection and development. This tool uses the available bioinformatics platform to assess antigenic variation within GAS genome sequences for any vaccine candidate of choice. This is an important step towards addressing current GAS diversity, in line with the vaccine pipeline [10].

# 4. Developments in Group A Streptococcus vaccines

Although GAS pathogens remain sensitive to penicillin, recent developments regarding the emergence of new GAS strains that are less sensitive to penicillin derivatives is alarming [8]. Hence, developing a safe, effective and affordable vaccine would significantly reduce the burden on human health from GAS disease and eliminate one of the top 10 infectious diseases worldwide. In the following section, we summarize the recent developments of GAS vaccine candidates in pre-clinical and clinical trials.

## 4.1. Group A Streptococcus vaccine candidates in clinical trials

To date, only two GAS vaccine candidates have completed human trials [84,85]. This includes the cell wall-anchored M protein, which has been the major focus in GAS vaccine research for several decades. Historically, the M protein has been known to trigger bactericidal antibodies that persist long after GAS infection in human serum. Reduced GAS colonization was initially observed in a 1970s study using administration of whole crude M protein extracts in human subjects. Therefore, despite their high-sequence diversity, M proteins are viewed as strong candidates for GAS vaccines [86]. However, due to the possible links of the M proteins with ARF, GAS vaccine development for humans was banned until 2004 [80]. Importantly, it was shortly revealed that M protein vaccine candidates were safe and protected against several GAS serotypes in the rodent infection model. The N-terminal hypervariable region and the C-terminal conserved region can elicit bactericidal antibodies. The N-terminal regions were formulated with recombinant proteins to produce the GAS

vaccine candidates: (i) 6-valent, (ii) 26-valent, and (iii) 30-valent. A synthetic peptide vaccine candidate from the C- terminal region was formulated to produce J8 M protein preparations completed the clinical trials (table 1).

Initial studies on M protein-based vaccine candidates focused on the N-terminal fragments of six different M proteins as antigens. These vaccine candidates were evaluated for their safety and immune response in a limited number of healthy volunteers. Notably, this 6-valent vaccine candidate did not cause any adverse effects. Built on this clinical study, the 26-valent vaccine candidate (StreptAvax) was developed and evaluated. It showed a positive impact on the overall burden of streptococcal infections in humans by reducing pharyngitis, NF and other streptococcal-related infections [89]. Even though no cross-reactive antibodies were identified, and the vaccine candidate tested negative for rheumatogenicity or nephritogenicity, StreptAvax failed in the major objective of inducing protection against a broad range of GAS strains. Notably, the StreptAvax vaccine candidate appeared to specifically target GAS strains commonly found in the Western countries and failed to offer protection against the GAS strains found in Asia and the Pacific continents [109]. The processing of StreptAvax was stopped due to commercial reasons [84].

The 30-valent vaccine candidate (StreptAnova™) was designed using N-terminal peptides from 30 M proteins. This vaccine candidate evoked opsonic antibody production protecting against GAS infections in rabbits [110], although the protection has not been firmly established using animal challenge models. In pre-clinical trials, these opsonic antibodies raised from rabbits killed not only the GAS isolates used to develop the vaccine, but also non-vaccine serotypes. An assessment of greater than 40% bactericidal activity (24/40) was observed against an arbitrary selection of non-vaccine serotypes [110]. This indicates broad coverage and neutralizing ability beyond the M protein serotypes. In the clinical phase I study, no evidence of autoimmunity or cross-reactive antibodies was recorded in 23 participants [84]. However, a different study pointed out safety concerns regarding regions with a high incidence of RHD [18]. Importantly, the 30-valent vaccine only achieves approximately 33% of antigenic coverage within vaccine targets from the 2083 GAS genomes [10].

An additional M protein domain from the C-terminal region was investigated as a vaccine candidate: J8 (MJ8VAX) contains a 29-amino acid peptide sequence from the C terminus of the M protein conjugated to the carrier protein. Analogous to the aforementioned vaccine candidates, J8 stimulated the production of opsonic antibodies in animals, which correlated with protection against intranasal and intraperitoneal GAS infection models [111,112]. J8-mediated protection was also shown in mice in conjunction with the SpyCEP immunogenic fragment [113]. A double-blinded, randomized phase I clinical trial demonstrated MJ8VAX to be safe and immunogenic [85], but, although the adverse effects were classified as mild in all ten participants, the level of antibodies decreased with time. The effectiveness of the MJ8VAX vaccine candidate was judged to be erratic, based on the low number of participants.

## 4.2. Group A Streptococcus vaccine candidates in the pre-clinical phase

Several vaccine candidates are currently in pre-clinical trials, and we highlight those that have shown protective immunity in animals. It is worthwhile mentioning that, up to now, none of the non-M-based vaccine candidates has reached clinical trials. The vaccine candidates in pre-clinical trials include non-M antigens such as streptococcal pyrogenic exotoxin, streptococcal C5a peptidase, streptolysin O, group A carbohydrate and derivatives (delta-GlcNAc-GAC/polyrhamnose), streptococcal pyrogenic exotoxin B, fibronectin-binding protein SfbI, *Streptococcus pyogenes* cell envelope proteinase, arginine deiminase and serum opacity factor. Published data is available for all candidates, revealing insights into their potency against GAS infections in animal studies (table 1).

### 4.2.1. Streptococcal C5a peptidase

The ScpA, a large surface multi-domain protein expressed on the cell envelope of GAS, has 98% amino acid sequence identity among the tested GAS serotypes. ScpA is one of the vital mediators of resistance to phagocytosis by specifically cleaving the chemotaxin C5a from the surface of polymorphonuclear leucocytes. In children with acute pharyngitis, GAS isolates from throat swabs revealed that ScpA was highly immunogenic, producing antibodies in the convalescent sera collected four weeks after infection [25]. Immunization of mice with ScpA mutants produced high titres of IgG1 and T cell populations. Furthermore, ScpA conjugated with short synthetic polysaccharides of the GAC was shown to trigger highly active T cell-dependent populations, suggesting that ScpA could be considered as a carrier protein in formulating 'combinational' GAS carbohydrate-based vaccines [114].

**Table 1.** List of GAS vaccine candidates in clinical and pre-clinical trials. I.M., intramuscular; I.P., intraperitoneal; S.C., subcutaneous; I.N., intranasal.

| vaccine candidates | description | outcome | reference |
|---|---|---|---|
| **clinical trials** | | | |
| 6-valent vaccine | comprised N-terminal M protein fragments from serotypes M1, M3, M5, M6, M19 and M24<br>Phase I—immunized 28 healthy adults<br>clinical assessment, serological responses and cross-reactive antibodies were examined post-vaccination | no tissue cross-reactive antibodies<br>30% increase in serum bactericidal activity post-vaccination<br>first evidence in humans that a multi-component protein elicits opsonic antibodies against GAS | [87,88] |
| 26-valent vaccine (StreptAvax) | comprised four recombinant proteins containing N-terminal peptides from 26 M proteins<br>Phase I—immunized 30 healthy adults<br>Phase II—immunized 30 healthy adults | the absence of rheumatogenicity or nephritogenicity<br>no induction of human tissue-reactive antibodies<br>a fourfold increase of IgG compared with control<br>discontinued due to commercial reasons | [89–91] |
| 30-valent vaccine (StreptAnova™) | comprised four recombinant proteins containing N-terminal peptides from 30 M proteins<br>Phase I—immunized 23 healthy adults | no evidence of autoimmunity<br>no tissue cross-reactive antibodies<br>25 out of 31 M serotypes showed significant antibody titre | [84] |
| J8 vaccine (MJ8VAX) | comprised a synthesized and acetylated peptide antigen (J8) from the conserved carboxyl terminus region of the M protein<br>Phase I—immunized 10 healthy adults | 13 adverse effects were classified as 'mild'<br>highly immunogenic after post-immunization<br>level of antibodies decreased with time | [85,92] |
| **pre-clinical trails** | | | |
| serum opacity factor (SOF) | function: opacifies mammalian serum<br>anti-SOF antibodies tested against M2, M4 and M28 | SOF stimulates antibodies in humans, rabbits and mice<br>provokes protective immunity by killing M4 and M28 | [93–95] |
| Group A carbohydrate (GAC) | GAC without GlcNAc side chain (polyrhamnose) used as an immunogen<br>purified GlcNAc-deficient GAC was tested for GAS survival<br>GlcNAc linked to the pathogenesis of rheumatic carditis | GlcNAc-deficient GAC facilitates opsonization and phagocytosis of diverse GAS strains<br>protects systemic and nasal challenges on mice and rabbit models<br>GlcNAc promotes GAS survival in human blood | [29,96,97] |

(*Continued.*)

| vaccine candidates | description | outcome | reference |
|---|---|---|---|
| C5a peptidase (ScpA) | highly specific endopeptidase<br>major virulence factor anchored on the surface of GAS<br>samples collected from children infected with pharyngitis | children with pharyngitis had increased ScpA activity<br>level of ScpA correlates with an increased level of anti-SLO and anti-DNase B activity | [25,98] |
| pyrogenic exotoxins (Spe) | SpeA and SpeC superantigen<br>Spe linked to STSS | toxoids of SpeA stimulates protective antibody response<br>anti-superantigen antibodies protects mice from GAS nasopharynx infection<br>induction of variable β-specific T cells promotes GAS colonization | [24,99–101] |
| streptolysin O (SLO) | pore-forming toxin produced by GAS<br>animals immunized S.C. and challenged with GAS | inactivated SLO mutant animals exhibited decreased mortality compared with wild-type GAS<br>SLO mutant protects animals from lethal M1 challenge | [26,102] |
| chemokine cleaving protease (SpyCEP) | SpyCEP is expressed on the GAS surface and secreted<br>function cleaves IL-8<br>SpyCEP expression upregulated in NF<br>mice immunized I.M. with SpyCEP and challenged with GAS through I.M. and I.N. | reduced bacterial dissemination found in both GAS and *S. equi*<br>offers protection against other streptococcal species | [27,103,104] |
| SfbI and FBP54 | SfbI and FBP54 is a fibronectin-binding protein<br>plays a key role in bacterial attachment to host cell<br>SfbI—animals immunized I.N. and challenged with M23 and blood isolate NS239<br>FBP54—mice immunized either S.C or orally and challenged I.P. with GAS | SfbI vaccinated animals show 80% antibody efficacy homologous challenge and 90% in heterologous challenge<br>FBP54 immunized mice survived significantly longer following GAS challenges | [28,105] |
| **multi-component vaccines** | | | |
| Spy7 | comprised highly conserved streptococcal surface antigen expressed in *E. coli*<br>mice were immunized, and I.M. challenged with M1, M3, M12 and M89 | production of anti-streptococcal antibodies<br>limited the dissemination of M1 and M3 | [70] |

(*Continued.*)

| vaccine candidates | description | outcome | reference |
|---|---|---|---|
| three technologies | comprised Streptolysin O, Spy0269 and SpyCEP<br>mice immunized I.P. and challenged I.N. or I.P. with GAS | broad protective antibody response against M1, M6, M12 and M23<br>antibody-mediated GAS killing—classical whole blood bactericidal assay | [106] |
| combination vaccines | three combination vaccines were formulated<br>I—comprised SLO, IL-8, SpyCEP, ScpA, ADI and trigger factor<br>II—comprised conserved M protein-derived J8 peptide conjugated to ADI<br>III—GAC without N-GlcNAc<br>mice immunized and challenged S.C. with GAS<br>M1 protein was used as a positive control | all experimental vaccine candidates elicited antigen-specific antibody coupled with bactericidal activity<br>only positive control provided protection against S.C invasive disease model | [107] |
| Combo vaccine (Combo5) | comprised SLO, ADI, ScpA, SpyCEP and trigger factor<br>Indian rhesus macaques immunized I.M. and challenged I.N with GAS | Combo5 immunization induced antigen-specific IgG in rhesus macaques<br>IgG against Combo5 bind to live GAS but do not promote killing by HL-60 cells<br>decreased severity of clinical signs but not colonization in pharyngitis infection model<br>following work highlighted that using adjuvants containing saponin QS21 with antigens ADI, SpyCEP, ScpA, SLO and trigger factor resulted in significant protection against GAS invasive infection | [33,108] |
| 5CP | comprised sortase A, streptococcal C5a peptidase, *S. pyogenes* the adhesion and division protein, a fragment of SpyCEP (CEP-5) and streptolysin O<br>mice immunized with 5CP and challenged I.N. with GAS to study mucosal and systemic infection<br>mice immunized I.N. with 5CP and challenged S.C with GAS for skin abscess model | 5CP induced Th17 responses in the spleen of animals<br>Th17 responses induced by 5CP resolve more rapidly than induced by GAS suggesting competent Th17 response towards 5CP | [34] |

### 4.2.2. Streptolysin O

SLO is a secreted GAS pore-forming toxin found to be upregulated in virulent M1T1 GAS isolates and other leading genotypes. SLO promotes GAS resistance to phagocytosis in the human immune system [115]. It also modulates the function of neutrophils by suppressing the neutrophil oxidative burst and impairing the direction of migration, thereby helping the pathogen to survive within the host bloodstream. High titres of IgG, IgM and anti-SLO antibodies were found in the mouse models immunized with a mutant (non-pore-forming) form of streptolysin O, implying passive protection in GAS-infected animals. This study suggested the importance of the SLO toxoid in multi-component vaccine formulation, because of its ability to reverse the neutrophil function and induce protective immunity against lethal GAS challenge [26]. SLO is a highly conserved protein achieving 99% theoretical coverage of all GAS isolates [10].

### 4.2.3. Group A Carbohydrate

Group A Carbohydrate (GAC) is 100% conserved in all isolated serotypes and is composed of a polyrhamnose backbone with an immunodominant GlcNAc side-chain decorated with a negatively charged glycerol phosphate [29]. It is an abundant and essential component of the GAS cell and makes up around 50%, by weight, of the cell wall [29]. From the early 1990s, affinity-purified anti-GAC antibodies have been recognized for opsonizing M3, M6, M14 and M28 serotypes [116]. Follow-up studies demonstrated that animals immunized with GAC produced a protective immune response against systemic GAS challenges [96]. However, the GlcNAc side chain of GAC was shown to cross-react with cardiac myosin proteins, thereby stimulating autoimmune antibodies, which are a primary agent in causing RHD [117]. As a result, only the polyrhamnose backbone of the GAC was taken forward as a possible vaccine candidate. An extensive study, by van Sorge *et al.*, using mouse and rabbit GAS infection models revealed that GAC antibodies raised against the polyrhamnose backbone promoted opsonophagocytic killing of multiple GAS serotypes, suggesting the GAC backbone could be viewed as a potential universal vaccine for GAS infection [29].

### 4.2.4. Streptococcal pyrogenic exotoxins

Spe are a family of GAS-secreted extracellular toxins including SpeA and SpeC. The expression of Spe exotoxins is responsible for producing the prominent rash and 'strawberry' tongue during scarlet fever by stimulating the production of inflammatory cytokines. Exotoxins SpeA and SpeC, also called superantigens, are commonly isolated from STSS patients. Accumulating evidence suggests that, similar to SpeA and SpeC, SpeF also acts as superantigen by interacting with major histocompatibility complex class II molecules on antigen presenting cells and is involved in the activation of T cells [99,118]. Such activated T cells release enormous amounts of pro-inflammatory cytokines, such as IFN gamma and tumour-necrosis factor α, leading to fever and shock [119]. One of the observed characteristics of superantigens is their ability to induce an inflammatory Th1 response rather than a regulatory Th2 cytokine response. This has been observed in STSS patients and implies a vital role for this in controlling the severity of invasive infections [120]. Conversely, SpeB, a cysteine protease, cleaves and degrades host proteins such as fibronectin and vitronectin and converts IL-1$\beta$ into an active molecule. Increased production of anti-SpeB antibodies was found in the sera of a diverse range of invasive disease patients [121]. Immunization with attenuated SpeA toxoid in the murine model of GAS nasopharyngeal infection resulted in antibody-mediated protection by reducing the burden of GAS in the nasopharynx, verifying superantigen SpeA as an attractive vaccine candidate for GAS [24].

### 4.2.5. Multi-component vaccines

A promising GAS vaccine candidate should provide broad coverage and protection against various serotypes, including emerging new GAS strains. Thus, developing a multi-component vaccine would be the best solution as it could potentially deliver protection based on factors such as antigenic variation, high immunogenicity and sequence conservation in most GAS serotypes. For example, the Combo5 vaccine candidate, a multi-component vaccine composed of GAS antigens such as streptolysin O, streptococcal c5a peptidase, arginine deiminase, *Streptococcus pyogenes* cell envelope proteinase and trigger factor, was shown to reduce pharyngitis and tonsillitis in Indian rhesus macaques, post-vaccination [33]. Moreover, subsequent work indicated that Combo5 with adjuvants containing saponin QS21 stimulated a Th1-type response, highlighting the importance of adjuvants

for designing non-M protein-based vaccine candidates for GAS [108]. A different study investigated a similar multi-component vaccine, termed 5CP. Intranasal immunization studies in mice revealed that 5CP not only stimulates T helper type 17 cells but also resolves them promptly to avoid Th17-induced autoimmune disorders, indicating a controlling role in the Th17 response [34]. Further studies investigated the potential of seven-GAS antigens as a combo vaccine candidate, spy7. Spy7 contains a variety of highly conserved streptococcal surface antigens that were recombinantly expressed in *E. coli*. All targets were purified and mixed in equal proportion to formulate the multi-component vaccine candidate. Spy7 vaccination in mice produced anti-streptococcal antibodies that prevented systemic dissemination of M1 and M3 GAS [70].

# 5. Concluding remarks

During the last 60 years, significant advances have been made in our understanding of GAS pathogenesis and disease, with detailed insights into the molecular mechanisms of pathogenicity including the ability of GAS to eradicate the host immune system and cause invasive infections. Invaluable information was generated through extensive genome analysis and evaluation of various GAS antigens as vaccine candidates. Numerous clinical trials and animal studies have been carried out to address how a single antigenic virulence determinant, such as the M proteins, can resist major innate immune players, including macrophages, neutrophils and dendritic cells, and cause serious invasive infections. Growing evidence for GAS antibiotic resistance, and the high burden of GAS on mankind, contributed to the WHO's decision to make GAS vaccine development a priority for global health. The only way forward to address the vast diversity of GAS isolates is to develop a universal vaccine candidate using combinational antigenic determinants, including proteins and carbohydrate glycoconjugates. A systematic approach that addresses multiple antigenic GAS components must be applied to deliver broad coverage of clinical infections, including careful analysis to avoid any potential cross-reactivity. The GAS research community has provided promising avenues to succeed and make a positive and lasting impact on global health.

# Abbreviations

Group A Streptococcus, GAS; necrotizing fasciitis, NF; streptococcal toxic shock syndrome, STSS; acute rheumatic fever, ARF; rheumatic heart disease, RHD; World Health Organization, WHO; streptococcal pyrogenic exotoxin, Spe; fibronectin-binding proteins, FBI; *Streptococcus pyogenes* cell envelope proteinase, SpyCEP; N-acetylglucosamine, GlcNAc; *Streptococcus pyogenes* fibronectin-binding adhesin, SfbI; streptococcal C5a peptidase, ScpA; streptolysin, O-SLO; extracellular matrix, ECM; hyaluronic acid capsule, HA; *Streptococcus pyogenes* adhesion and division protein, spyAD; Group A Carbohydrate, GAC.

Data accessibility. This article does not contain any additional data.
Authors' contributions. S.A.C. and H.C.D. wrote the review article. S.A.C. designed figures and compiled table.
Competing interests. We declare we have no competing interests.
Funding. S.A.C. is funded by Tenovus Scotland Large Research Grant (T17/17) and The Wellcome Trust (105606/Z/14/Z). H.C.D. is funded by the Royal Society and The Wellcome Trust Sir Henry Dale Fellowship (109357/Z/15/Z).
Acknowledgement. We would like to thank Dr Azul Zorzoli and Dr Mark Reglinski for critical reading of the review.

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
