## [Peer Review File · Royal Society Open Science]

Review History

RSOS-201991.R0 (Original submission)

Review form: Reviewer 1

Is the manuscript scientifically sound in its present form?

Yes

Are the interpretations and conclusions justified by the results?

Yes

Is the language acceptable?

Yes

Do you have any ethical concerns with this paper?

No

Have you any concerns about statistical analyses in this paper?

No

Recommendation?

Accept with minor revision (please list in comments)

Comments to the Author(s)

I am very happy about this review overall. It is informative and well written.

I do have a few minor points I would like to highlight, though.

- 1) The text contains many abbreviations. Would it be possible to decrease their number to make the text easier to read? Also, most of them are not present in the Abbreviation section. It would be very helpful if the full list was accessible there.
- 2) Could you add a figure representing the M protein? It is often mentioned in the text so I think it would be worth it to better visualise what is known about it (structure, domains used in candidate vaccines, etc).
- 3) A few sentences are very long and thus difficult to read.
 - a. Line 70 : end the sentence after Amoxicillin and start a new one with "The most promising..."
 - b. Line 104-108 : can you break them into several smaller sentences?
 - c. Line 102: start a new sentence at "However".
 - d. Line 320-325 : replace by "...in scarlet fever [63], in a nursing facility... multiple invasive GAS infections [64], and recently in 2019, Public ..."
 - e. Line 512-514 : replace by "stimulating the production of inflammatory cytokines. SpeA and SpeC exotoxins, also considered as superantigens, are..."
- 4) A few references are missing. I guess the next given reference applies to the preceding sentences, but it is not always obvious. Here are a few lines where a clearer reference would be helpful.
 - a. Line 51
 - b. Line 97
 - c. Line 116
 - d. Line 145
 - e. Line 155
 - f. Line 159
 - g. Line 305
- 5) Other comments :
 - a. Line 73 : "Production" should not have a capital letter
 - b. Line 347 : "includes" should not have a terminal S
 - c. Line 523 : add a comma after "cysteine protease"

Review form: Reviewer 2

Is the manuscript scientifically sound in its present form?

No

Are the interpretations and conclusions justified by the results?

No

Is the language acceptable?

No

Do you have any ethical concerns with this paper?

No

Have you any concerns about statistical analyses in this paper?

No

Recommendation?

Major revision is needed (please make suggestions in comments)

Comments to the Author(s)

The authors have written a comprehensive review of the group A Streptococcus (GAS) field, with focus on vaccine development. However, the manuscript itself should be proof read by an editorial specialist as it contains less than ideal English in sections.

Comments for the authors to address:

1. Page 5. Note that a brief review has recently been published describing progress of M protein and non-M protein GAS vaccine development. *Current Opinion in Infectious Diseases* (2020) 33: 244-250.

Line 184 "Genes of the component regulatory system [CovRS]". Please articulate that "Cov" stands for "Control of Virulence".

Line 209 The use of caveolae by GAS for uptake into epithelial cells is not conclusive. See following manuscript: Caveolin-1 restricts group A Streptococcus invasion of non-phagocytic host cells. *Cellular Microbiology* 19: e12772.

Lines 227-8. It is not established that invasion of epithelial cells directly contributes to invasive disease. E.g. See for example recent paper from Sriskandan lab showing the systemic transmission of GAS through the lymphatic system. Extracellular bacterial lymphatic metastasis drives Streptococcus pyogenes systemic infection
MK Siggins et al. *Nature communications* 11, 1-12.

Lines 230-236. It is not clear what the authors are trying to say here. These are very old studies, and the field has largely moved on from these hypotheses. Again, no connection proven between intracellular invasion and GAS invasive disease.

Lines 247-256. Note that China/Hong Kong scarlet fever isolates have association with SSA, SpeC and Spd1.

Lines 265-266. This suggests that both the HA capsule and the M protein, present in almost all 266 GAS strains, contribute to the deadly NF disease. This statement is not correct. New outbreak strains of M89 and M4 strains of GAS causing invasive infections do not produce capsule.

Line 273. Replace Western regions with Western countries

Lines 278-279. In association with this, a few studies have noted a link between the 279 M1, M3 strains and SpeA expression and STSS [54, 55]. Note this observations is a correlation, not causation.

Line 316. acute post-streptococcal glomerulonephritis [ASPGN], a prototype of PSGN. this line has no meaning. APSGN is not a prototype.

Line 317. V1 region in streptokinase plasminogen. Streptokinase is not plasminogen, nor vice versa. They are 2 different proteins that interact.

Line 332-334. Not a single protein that has been exploited for vaccine development so far, has been identified being 100% conserved throughout all GAS isolates [10]. What does this even mean?

Line 403. To date only two GAS vaccine candidates are currently in clinical trials. Wrong - 2 candidates have completed human trials.

Line 429-431. Notably, the StreptAvax vaccine candidate appears to benefit infections caused by GAS strains commonly found in the Western countries... This is grammatically incorrect.

Line 434-435. In fact, there was no challenge study using StreptAvax reported in this reference - only OPK assays.

Lines 435-437. In pre-clinical trials, these opsonic antibodies raised from rabbits killed not only the GAS isolates used to develop the vaccine, but also the non-vaccine serotypes. Not all serotypes, only some others, and an arbitrary 40% cut off was used in any case. Bactericidal activity of >40% was observed against 24/40 serotypes.

Lines 441-443. Importantly, the 30-valent vaccine only achieves approximately 33% of antigenic variation within vaccine targets from the 2083 GAS genomes [10]. Please check accuracy. I think you need to change "variation" for "coverage".

Lines 457-458. It is worthwhile mentioning that, until today, none of the non-M-based vaccine candidates have approached clinical trials. I think you need to change "approached" for "reached".

Lines 523-524. Conversely, SpeB, a cysteine protease cleaves and degrades host proteins such as fibronectin, vitronectin and IL-1 β into active molecule. My understanding is the SpeB converts only IL-1B into an active molecule.

Lines 537-540. 'Combo vaccine [5CP]', a multicomponent vaccine composed of GAS antigens such as SLO, ScpA, sortase A [srtA], SpyCEP, and Streptococcus pyogenes adhesion and division protein [spyAD] was shown to reduce pharyngitis and tonsillitis in Indian rhesus macaque post vaccination [24].

Lines 537-540. In reference 24, the antigens used were SLO, SpyCEP, ScpA, ADI and TF. Please cross-reference the original manuscript.

Lines 540-542. This is a different vaccine being referred to. The text reads as if you are discussing same vaccine.

Page 38 Under "Combination vaccines". This section is very confused, as you have listed SLO, SpyCEP, ScpA, ADI, J8 and GAC without NGlcNaC. These are referring to THREE different vaccines, not a single combination. Suggest you go back and read the paper, then accurately summarise the results in your Table. Also note that a follow up manuscript showed that using new adjuvants in combination with the SLO, ScpA, SpyCEP, ADI and TF antigens, this new combination vaccine resulted in significant protection in the same invasive disease model. See mBio 11: e00122-20. .

Decision letter (RSOS-201991.R0)

This year has been very difficult for everyone, and we want to take the opportunity to thank you for your continued support in 2020.

The Royal Society Open Science editorial office will be closed from the evening of Friday 18 December 2020 until Monday 4 January 2021. We will not be responding during this time. If you have received a deadline within this time period, please contact us as soon as possible to allow us to extend the deadline. If you receive any automated messages during this time asking you to meet a deadline, we offer apologies and invite you to respond after the festive period or during normal working hours.

With our best for a peaceful festive period and New Year, and we look forward to working with you in 2021.

Dear Dr Dorfmueller

On behalf of the Editors, we are pleased to inform you that your Manuscript RSOS-201991 "A Brief Review on Group A Streptococcus Pathogenesis and Vaccine Development" has been accepted for publication in Royal Society Open Science subject to minor revision in accordance with the referees' reports. Please find the referees' comments along with any feedback from the Editors below my signature. As you will see, the manuscript requires significant textual revisions before re-submission.

Please submit your revised manuscript and required files (see below) no later than 7 days from today's (ie 18-Dec-2020) date. Note: the ScholarOne system will 'lock' if submission of the revision is attempted 7 or more days after the deadline. If you do not think you will be able to meet this deadline please contact the editorial office immediately.

on behalf of Dr Shaked Ashkenazi (Associate Editor) and Malcolm White (Subject Editor)
openscience@royalsociety.org

Reviewer comments to Author:

Reviewer: 1

Comments to the Author(s)

I am very happy about this review overall. It is informative and well written.

I do have a few minor points I would like to highlight, though.

1) The text contains many abbreviations. Would it be possible to decrease their number to make the text easier to read? Also, most of them are not present in the Abbreviation section. It would be very helpful if the full list was accessible there.

2) Could you add a figure representing the M protein? It is often mentioned in the text so I think it would be worth it to better visualise what is known about it (structure, domains used in candidate vaccines, etc).

3) A few sentences are very long and thus difficult to read.

a. Line 70 : end the sentence after Amoxicillin and start a new one with "The most promising..."

b. Line 104-108 : can you break them into several smaller sentences?

c. Line 102: start a new sentence at "However".

d. Line 320-325 : replace by "...in scarlet fever [63], in a nursing facility... multiple invasive GAS infections [64], and recently in 2019, Public ..."

e. Line 512-514 : replace by "stimulating the production of inflammatory cytokines. SpeA and SpeC exotoxins, also considered as superantigens, are..."

4) A few references are missing. I guess the next given reference applies to the preceding sentences, but it is not always obvious. Here are a few lines where a clearer reference would be helpful.

a. Line 51

b. Line 97

c. Line 116

d. Line 145

e. Line 155

f. Line 159

g. Line 305

5) Other comments :

a. Line 73 : "Production" should not have a capital letter

b. Line 347 : "includes" should not have a terminal S

c. Line 523 : add a comma after "cysteine protease"

Reviewer: 2

Comments to the Author(s)

The authors have written a comprehensive review of the group A Streptococcus (GAS) field, with focus on vaccine development. However, the manuscript itself should be proof read carefully as it contains less than ideal English in sections.

Comments for the authors to address:

1. Page 5. Note that a brief review has recently been published describing progress of M protein and non-M protein GAS vaccine development. Current Opinion in Infectious Diseases (2020) 33: 244-250.

Line 184 "Genes of the component regulatory system [CovRS]". Please articulate that "Cov" stands for "Control of Virulence".

Line 209 The use of caveolae by GAS for uptake into epithelial cells is not conclusive. See following manuscript: Caveolin-1 restricts group A Streptococcus invasion of non-phagocytic host cells. Cellular Microbiology 19: e12772.

Lines 227-8. It is not at all established that invasion of epithelial cells directly contributes to invasive disease. E.g. See for example recent paper from Sriskandan lab showing the systemic transmission of GAS through the lymphatic system. Extracellular bacterial lymphatic metastasis drives Streptococcus pyogenes systemic infection
MK Siggins et al. Nature communications 11, 1-12.

Lines 230-236. It is not clear what the authors are trying to say here. These are very old studies, and the field has largely moved on from these hypotheses. Again, no connection proven between intracellular invasion and GAS invasive disease.

Lines 247-256. Note that China/Hong Kong scarlet fever isolates have association with SSA, SpeC and Spd1.

Lines 265-266. This suggests that both the HA capsule and the M protein, present in almost all 266 GAS strains, contribute to the deadly NF disease. This statement is not correct. New outbreak strains of M89 and M4 strains of GAS causing invasive infections do not produce capsule.

Line 273. Replace Western regions with Western countries

Lines 278-279. In association with this, a few studies have noted a link between the 279 M1, M3 strains and SpeA expression and STSS [54, 55]. Note this observations is a correlation, not causation.

Line 316. acute post-streptococcal glomerulonephritis [ASPGN], a prototype of PSGN. this line has no meaning. APSGN is not a prototype.

Line 317. V1 region in streptokinase plasminogen. Streptokinase is not plasminogen, nor vice versa. They are 2 different proteins that interact.

Line 332-334. Not a single protein that has been exploited for vaccine development so far, has been identified being 100% conserved throughout all GAS isolates [10]. Please clarify this sentence.

Line 403. To date only two GAS vaccine candidates are currently in clinical trials. In fact - 2 candidates have completed human trials; please revise accordingly

Line 429-431. Notably, the StreptAvax vaccine candidate appears to benefit infections caused by GAS strains commonly found in the Western countries... This is gramatically incorrect, please revise

Line 434-435. In fact, these was no challenge study using StreptAvax reported in this reference - only OPK assays.

Lines 435-437. In pre-clinical trials, these opsonic antibodies raised from rabbits killed not only the GAS isolates used to develop the vaccine, but also the non-vaccine serotypes. Not all

serotypes, only some others, and an arbitrary 40% cut off was used in any case. Bactericidal activity of >40% was observed against 24/40 serotypes.

Lines 441-443. Importantly, the 30-valent vaccine only achieves approximately 33% of antigenic variation within vaccine targets from the 2083 GAS genomes [10]. Please check accuracy. I think you need to change "variation" for "coverage".

Lines 457-458. It is worthwhile mentioning that, until today, none of the non-M-based vaccine candidates have approached clinical trials. I think you need to change "approached" for "reached".

Lines 523-524. Conversely, SpeB, a cysteine protease cleaves and degrades host proteins such as fibronectin, vitronectin and IL-1 β into active molecule. My understanding is the SpeB converts only IL-1B into an active molecule.

Lines 537-540. 'Combo vaccine [5CP]', a multicomponent vaccine composed of GAS antigens such as SLO, ScpA, sortase A [srtA], SpyCEP, and Streptococcus pyogenes adhesion and division protein [spyAD] was shown to reduce pharyngitis and tonsillitis in Indian rhesus macaque post vaccination [24].

Lines 537-540. In reference 24, the antigens used were SLO, SpyCEP, ScpA, ADI and TF. Please cross-reference the original manuscript.

Lines 540-542. This is a different vaccine being referred to. The text reads as if you are discussing same vaccine.

Page 38 Under "Combination vaccines". This section is very confused, as you have listed SLO, SpyCEP, ScpA, ADI, J8 and GAC without NGlcNaC. These are referring to THREE different vaccines, not a single combination. Suggest you go back and read the paper, then accurately summarise the results in your Table. Also note that a follow up manuscript showed that using new adjuvants in combination with the SLO, ScpA, SpyCEP, ADI and TF antigens, this new combination vaccine resulted in significant protection in the same invasive disease model. See mBio 11: e00122-20. .

===PREPARING YOUR MANUSCRIPT===

While not essential, it will speed up the preparation of your manuscript proof if you format your references/bibliography in Vancouver style (please see

<https://royalsociety.org/journals/authors/author-guidelines/#formatting>). You should include DOIs for as many of the references as possible.

===PREPARING YOUR REVISION IN SCHOLARONE===

<https://royalsociety.org/journals/authors/author-guidelines/#data>. You should ensure that you cite the dataset in your reference list. If you have deposited data etc in the Dryad repository,

please only include the 'For publication' link at this stage. You should remove the 'For review' link.

Author's Response to Decision Letter for (RSOS-201991.R0)

See Appendix A.

Decision letter (RSOS-201991.R1)

Dear Dr Dorfmüller

On behalf of the Editors, we are pleased to inform you that your Manuscript RSOS-201991.R1 "A Brief Review on Group A Streptococcus Pathogenesis and Vaccine Development" has been accepted for publication in Royal Society Open Science subject to minor revision in accordance with the referees' reports. Please find the referees' comments along with any feedback from the Editors below my signature.

Please submit your revised manuscript and required files (see below) no later than 7 days from today's (ie 03-Feb-2021) date. Note: the ScholarOne system will 'lock' if submission of the revision is attempted 7 or more days after the deadline. If you do not think you will be able to meet this deadline please contact the editorial office immediately.

Please note article processing charges apply to papers accepted for publication in Royal Society Open Science (<https://royalsocietypublishing.org/rsos/charges>). Charges will also apply to papers transferred to the journal from other Royal Society Publishing journals, as well as papers submitted as part of our collaboration with the Royal Society of Chemistry

(<https://royalsocietypublishing.org/rsos/chemistry>). Fee waivers are available but must be requested when you submit your revision (<https://royalsocietypublishing.org/rsos/waivers>).

on behalf of Dr Shaked Ashkenazi (Associate Editor) and Malcolm White (Subject Editor)
openscience@royalsociety.org

Associate Editor Comments to Author (Dr Shaked Ashkenazi):

Associate Editor

Comments to the Author:

Thank you for thoroughly addressing the reviewers' comments and suggestions.

A significant part of the introduction is dedicated to describing the M protein, including the gene structure, different domains, etc. One of the reviewers asked for a schematic representation of the M protein, to which the authors replied that it is available elsewhere. While I acknowledge that this is not the focus of this review, I feel that the description of the M protein is too detailed to ignore that. i.e., How many repeat sequences, what is their nature, where is each one found with regards to the protein termini, etc. In my view, the reader should not be reading about the exact details of the gene structure without having a convenient illustration at hand, as this may be hard to follow. I would request that you either add an illustration or revise the paragraph so that it is easier to follow (perhaps some of the details can be spared?).

===PREPARING YOUR MANUSCRIPT===

===PREPARING YOUR REVISION IN SCHOLARONE===

Author's Response to Decision Letter for (RSOS-201991.R1)

See Appendix B.

Decision letter (RSOS-201991.R2)

Dear Dr Dorfmueller,

It is a pleasure to accept your manuscript entitled "A Brief Review on Group A Streptococcus Pathogenesis and Vaccine Development" in its current form for publication in Royal Society Open Science. The comments of the reviewer(s) who reviewed your manuscript are included at the foot of this letter.

Please see the Royal Society Publishing guidance on how you may share your accepted author manuscript at <https://royalsociety.org/journals/ethics-policies/media-embargo/>. After

publication, some additional ways to effectively promote your article can also be found here <https://royalsociety.org/blog/2020/07/promoting-your-latest-paper-and-tracking-your-results/>.

Kind regards,

Anita Kristiansen
Editorial Coordinator

on behalf of Dr Shaked Ashkenazi (Associate Editor) and Malcolm White (Subject Editor)
openscience@royalsociety.org

Associate Editor Comments to Author (Dr Shaked Ashkenazi):
Comments to the Author: All requested modifications made, thank you.

Appendix A

Reviewer: 1

Comments to the Author(s)

I am very happy about this review overall. It is informative and well written.

I do have a few minor points I would like to highlight, though.

1) The text contains many abbreviations. Would it be possible to decrease their number to make the text easier to read? Also, most of them are not present in the Abbreviation section. It would be very helpful if the full list was accessible there.

We agree that the number of abbreviations is relatively large. Unfortunately, we wouldn't be able to reduce the abbreviations as most of the abbreviations are used for table and figures. For example, if we write the full text in the table, it will not look appealing and accommodated. However, the abbreviation list has now been revised and updated, as suggested.

2) Could you add a figure representing the M protein ? It is often mentioned in the text so I think it would be worth it to better visualise what is known about it (structure, domains used in candidate vaccines, etc).

This review concentrates specifically on the current development of GAS vaccine candidates. We have noticed that a number of specific reviews have been written with a focus on the structure and mechanisms of M-proteins. Therefore, we have directed the readers to these excellent figures mentioned in the recent publications.

3) A few sentences are very long and thus difficult to read.

a. Line 70 : end the sentence after Amoxicillin and start a new one with "The most promising..."

b. Line 104-108 : can you break them into several smaller sentences?

c. Line 102: start a new sentence at "However".

d. Line 320-325 : replace by "...in scarlet fever [63], in a nursing facility... multiple invasive GAS infections [64], **and** recently in 2019, Public ..."

e. Line 512-514 : replace by "stimulating the production of inflammatory cytokines. **SpeA** and **SpeC** exotoxins, also considered as superantigens, are..."

We thank the reviewer for highlighting these sections and we have revised them to aid the reader.

4) A few references are missing. I guess the next given reference applies to the preceding sentences, but it is not always obvious. Here are a few lines where a clearer reference would be helpful.

a. Line 51, b. Line 97, c. Line 116, d. Line 145, e. Line 155, f. Line 159, g. Line 305

We have corrected the manuscript with regards to the suggested references and carefully checked all other references.

5) Other comments :

a. Line 73 : "Production" should not have a capital letter

b. Line 347 : "includes" should not have a terminal S

c. Line 523 : add a comma after “cysteine protease”

We have corrected these typos and a number of additional typos and grammatical mistakes.

Reviewer: 2

Comments to the Author(s)

The authors have written a comprehensive review of the group A Streptococcus (GAS) field, with focus on vaccine development. However, the manuscript itself should be proof read carefully as it contains less than ideal English in sections.

Comments for the authors to address:

1. Page 5. Note that a brief review has recently been published describing progress of M protein and non-M protein GAS vaccine development. Current Opinion in Infectious Diseases (2020) 33: 244-250.

We have incorporated this reference.

Line 184 "Genes of the component regulatory system [CovRS]". Please articulate that "Cov" stands for "Control of Virulence".

We have articulated this abbreviation accordingly.

Line 209 The use of caveolae by GAS for uptake into epithelial cells is not conclusive. See following manuscript: Caveolin-1 restricts group A Streptococcus invasion of non-phagocytic host cells. Cellular Microbiology 19: e12772.

We thank the reviewer to highlight this fact and we have edited the manuscript to reflect this.

Lines 227-8. It is not at all established that invasion of epithelial cells directly contributes to invasive disease. E.g. See for example recent paper from Sriskandan lab showing the systemic transmission of GAS through the lymphatic system. Extracellular bacterial lymphatic metastasis drives Streptococcus pyogenes systemic infection MK Siggins et al. Nature communications 11, 1-12.

We have included the reference and correct the sentence.

Lines 230-236. It is not clear what the authors are trying to say here. These are very old studies, and the field has largely moved on from these hypotheses. Again, no connection proven between intracellular invasion and GAS invasive disease.

As the reviewer pointed correctly out, that there is no proven connection, we have removed this paragraph.

Lines 247-256. Note that China/Hong Kong scarlet fever isolates have association with SSA, SpeC and Spd1.

We have revised the sentence accordingly.

Lines 265-266. This suggests that both the HA capsule and the M protein, present in almost all 266 GAS strains, contribute to the deadly NF disease. This statement is not correct. New outbreak strains of M89 and M4 strains of GAS causing invasive infections do not produce capsule.

We have corrected the sentence.

Line 273. Replace Western regions with Western countries

We have corrected this sentence.

Lines 278-279. In association with this, a few studies have noted a link between the 279 M1, M3 strains and SpeA expression and STSS [54, 55]. Note this observations is a correlation, not causation.

We have corrected this sentence.

Line 316. acute post-streptococcal glomerulonephritis [ASPGN], a prototype of PSGN. this line has no meaning. APSGN is not a prototype.

We have corrected this sentence.

Line 317. V1 region in streptokinase plasminogen. Streptokinase is not plasminogen, nor vice versa. They are 2 different proteins that interact.

Many thanks for pointing this out. We have corrected this.

Line 332-334. Not a single protein that has been exploited for vaccine development so far, has been identified being 100% conserved throughout all GAS isolates [10]. Please clarify this sentence.

We have revised and clarified this sentence.

Line 403. To date only two GAS vaccine candidates are currently in clinical trials. In fact - 2 candidates have completed human trials; please revise accordingly

We have corrected this sentence.

Line 429-431. Notably, the StreptAvax vaccine candidate appears to benefit infections caused by GAS strains commonly found in the Western countries... This is gramatically incorrect, please revise

We have grammatically corrected this sentence.

Line 434-435. In fact, these was no challenge study using StreptAvax reported in this reference - only OPK assays.

We have checked the literature and corrected this sentence.

Lines 435-437. In pre-clinical trials, these opsonic antibodies raised from rabbits killed not only

the GAS isolates used to develop the vaccine, but also the non-vaccine serotypes. Not all serotypes, only some others, and an arbitrary 40% cut off was used in any case. Bactericidal activity of >40% was observed against 24/40 serotypes.

We have corrected this sentence.

Lines 441-443. Importantly, the 30-valent vaccine only achieves approximately 33% of antigenic variation within vaccine targets from the 2083 GAS genomes [10]. Please check accuracy. I think you need to change "variation" for "coverage".

We have corrected this sentence.

Lines 457-458. It is worthwhile mentioning that, until today, none of the non-M-based vaccine candidates have approached clinical trials. I think you need to change "approached" for "reached".

We have revised this sentence.

Lines 523-524. Conversely, SpeB, a cysteine protease cleaves and degrades host proteins such as fibronectin, vitronectin and IL-1 β into active molecule. My understanding is the SpeB converts only IL-1B into an active molecule.

That understanding is correct and we have revised this sentence.

Lines 537-540. 'Combo vaccine [5CP]', a multicomponent vaccine composed of GAS antigens such as SLO, ScpA, sortase A [srtA], SpyCEP, and Streptococcus pyogenes adhesion and division protein [spyAD] was shown to reduce pharyngitis and tonsillitis in Indian rhesus macaque post vaccination [24].

Lines 537-540. In reference 24, the antigens used were SLO, SpyCEP, ScpA, ADI and TF. Please cross-reference the original manuscript.

Lines 540-542. This is a different vaccine being referred to. The text reads as if you are discussing same vaccine.

*The **three points** were all revised and we included the correct details in the table.*

Page 38 Under "Combination vaccines". This section is very confused, as you have listed SLO, SpyCEP, ScpA, ADI, J8 and GAC without NGlcNaC. These are referring to THREE different vaccines, not a single combination. Suggest you go back and read the paper, then accurately summarise the results in your Table. Also note that a follow up manuscript showed that using new adjuvants in combination with the SLO, ScpA, SpyCEP, ADI and TF antigens, this new combination vaccine resulted in significant protection in the same invasive disease model. See mBio 11: e00122-20. .

We have included this suggested reference and revised the section accordingly.

In addition, we have also asked a professional proof-reader to edit the manuscript.

We hope these revisions are all acceptable.

Kind regards,

Helge Dorfmueller

Appendix B

Dear Dr Shaked Ashkenazi,

Many thanks for your kind email and offering us to revise the review. Please find our revisions below.

Comments to the Author:

Thank you for thoroughly addressing the reviewers' comments and suggestions.

A significant part of the introduction is dedicated to describing the M protein, including the gene structure, different domains, etc. One of the reviewers asked for a schematic representation of the M protein, to which the authors replied that it is available elsewhere. While I acknowledge that this is not the focus of this review, I feel that the description of the M protein is too detailed to ignore that. i.e., How many repeat sequences, what is their nature, where is each one found with regards to the protein termini, etc. In my view, the reader should not be reading about the exact details of the gene structure without having a convenient illustration at hand, as this may be hard to follow. I would request that you either add an illustration or revise the paragraph so that it is easier to follow (perhaps some of the details can be spared?).

We agree with the comment that details could be spared. We have removed details about the domain structures and N- and C-terminal domains and point directly to the recent literature.

We hope these revisions are all acceptable.

Kind regards,

Helge Dorfmueller